# Safety and Efficacy of the Transaxillary Access for Minimally Invasive Mitral Valve Surgery—A Propensity Matched Competitive Analysis

**DOI:** 10.3390/medicina58121850

**Published:** 2022-12-15

**Authors:** Ali Taghizadeh-Waghefi, Sebastian Arzt, Veronica De Angelis, Jana Schiffarth, Asen Petrov, Matuš Tomko, Konstantin Alexiou, Klaus Matschke, Utz Kappert, Manuel Wilbring

**Affiliations:** 1Center for Minimally Invasive Cardiac Surgery, University Heart Center Dresden, 01307 Dresden, Germany; 2Department of Cardiac Anesthesiology, University Heart Center Dresden, 01307 Dresden, Germany

**Keywords:** mitral valve, minimally invasive mitral valve surgery, outcomes

## Abstract

*Background and Objectives*: Transaxillary access is a straightforward “single incision—direct vision” concept, based on a 5 cm skin incision in the right anterior axillary line. It is suitable for aortic, mitral and tricuspid surgery. The present study evaluates the hospital outcomes of the transaxillary access for isolated mitral valve surgery compared with full sternotomy. *Patients and Methods*: The final study group included 480 patients. A total of 160 consecutive transaxillary patients served as treatment group (MICS-MITRAL). Based on a multivariate logistic regression model including age, sex, body-mass-index, EuroScore II and LVEF, a 1:2 propensity matched control-group (*n* = 320) was generated out of 980 consecutive sternotomy patients. Redo surgeries, endocarditis or combined procedures were excluded. The mean age was 66.6 ± 10.6 years, 48.6% (*n* = 234) were female. EuroSCORE II averaged 1.98 ± 1.4%. *Results*: MICS-MITRAL had longer perfusion (88.7 ± 26.6 min vs. 68.7 ± 32.7 min; *p* < 0.01) and cross-clamp (64.4 ± 22.3 min vs. 49.7 ± 22.4 min; *p* < 0.01) times. This did not translate into longer procedure times (132 ± 31 min vs. 131 ± 46 min; *p* = 0.76). Both groups showed low rates of failed repair (MICS-MITRAL: *n* = 6/160; 3.75%; Sternotomy: *n* = 10/320; 3.1%; *p* = 0.31). MICS-MITRAL had lower transfusion rates (*p* ≤ 0.001), less re-exploration for bleeding (*p* = 0.04), shorter ventilation times (*p* = 0.02), shorter ICU-stay (*p* = 0.05), less postoperative hemofiltration (*p* < 0.01) compared to sternotomy patients. No difference was seen in the incidence of stroke (*p* = 0.47) and postoperative delirium (*p* = 0.89). Hospital mortality was significantly lower in MICS-MITRAL patients (0.0% vs. 3.4%; *p* = 0.02). *Conclusions*: The transaxillary access for MICS-MITRAL provides superior cosmetics and excellent clinical outcomes. It can be performed at least as safely and in the same time frame as conventional mitral surgery by sternotomy.

## 1. Introduction

Minimally invasive techniques for cardiac surgery are in greater demand than ever. Patient desire for less trauma, less pain, faster recovery as well as better cosmetics and the parallel, ongoing rise of catheter-based techniques caused a strong push into minimally invasive techniques. This trend is particularly true for aortic valve disease but even though for the mitral valve. Nonetheless, until now, minimally invasive surgery failed to prove better outcomes by means of mortality and stroke rate compared to the classic full sternotomy approach. Scientifically approved key benefits of minimally invasive Cardiac Surgery (MICS) include reduced postoperative bleeding, less frequent transfusion, faster postoperative recovery, shorter hospital stay, fewer wound infections, and cosmetic benefits of a smaller surgical wound [1,2,3,4,5,6,7]. With regard to that, there is a reason to follow minimally invasive developments.

Although minimally invasive mitral valve surgery is continuously promoted over the past decades, only 56% of patients presenting with isolated mitral valve disease actually do undergo minimally invasive mitral valve surgery (MICS-MITRAL) in Germany [8]. A reason for this lack of penetrance compared to minimally invasive aortic valve surgery might be a result of the increasing complexity of the supposed techniques. For example, the latest promoted technique was the “the non-rib-spreading, fully 3D endoscopic mini-incision mitral valve surgery”, which undoubted represents a pivotal stage in the evolutionary development of isolated MICS-MITRAL [9,10]. However, this approach is associated with substantial logistical and financial requirements, and is suffering from a high case load to achieve and obtain the result quality of the individual surgeon [11]. Consequently, the procedure often is reserved for rarely more than one surgeon per institution, which at the end prevents its widespread use.

A simplified and easy-to-adopt alternative might be the transaxillary access which has been reported previously in the context of minimally invasive isolated or combined aortic and mitral valve surgery [12,13]. This “one incision—direct vision” access route is based on a 5 cm skin incision in the anterior axillary line to access the 3rd or 4th intercostal space. Through this access, surgery at the aortic, mitral or tricuspid valve is possible—without the use of sophisticated technical equipment [12,13]. Additionally, the cosmetic result is unbeatable, with a small scar disappearing near the right axilla [12,13] (Figure 1).

Nonetheless, this new and technically less demanding approach has not yet proven its safety and efficacy based on a large number of patients compared to full sternotomy as the gold standard in concerns of mortality and stroke.

For those purposes, the present study was set up. In this series, transaxillary access for minimally invasive mitral valve surgery is challenged by a propensity matched cohort of isolated mitral valve surgeries done by sternotomy.

## 2. Patients and Methods

### 2.1. Inclusion and Exclusion Criteria

The present study aimed for adult patients undergoing isolated mitral valve surgery using either the transaxillary access in the treatment group (MICS-MITRAL), or the full sternotomy in the conventionally treated control-group. Allowed concomitant procedures were closure of the left atrial appendage, closure of a persisting foramen ovale or ablation for atrial fibrillation. Exclusion criteria were other combined procedures, active or recent endocarditis or redo surgeries.

### 2.2. Study Design and Ethical Statement

This study is a single-center, retrospective cohort analysis of consecutive patients undergoing minimally invasive mitral valve surgery using the transaxillary access. For comparison, a 1:2 propensity score matching was performed using a logistic regression model. By doing this, the control-group was generated out of a retrospectively analyzed cohort of consecutive patients undergoing isolated mitral valve surgery by full sternotomy. For both groups patients were enrolled according to the above mentioned inclusion and exclusion criteria. The data were retrospectively obtained from the hospitals database. The study was reviewed and approved by the local Ethic Board.

### 2.3. Patients and Groups

The final study group consisted of 480 Patients. The mean age was 66.6 ± 10.6 years, sex was nearly balanced with 48.6% (*n* = 234) being female. The EuroSCORE II averaged 1.98 ± 1.4%, indicating a low-risk population.

The groups were generated out of 1.437 patients undergoing isolated mitral valve surgery according to the above mentioned inclusion and exclusion criteria in the time period between January 2000 and December 2021.

Transaxillary access consequently was introduced at our institution in 2019. Subsequently, all consecutive patients undergoing isolated mitral valve through the transaxillary access surgery fulfilling the inclusion and exclusion criteria until December 2021 were enrolled, and formed the treatment group. During the period between 2019 and 2021 a total of 168 patients underwent isolated mitral surgery according to the inclusion criteria. Hence, the corresponding MICS-proportion was 95.2%. Eight patients (4.8%) were not done transaxillary, due to extensive annular calcifications with conceivable need for decalcification and possible need for posterior mitral patch annuloplasty.

The control-group of patients undergoing mitral valve surgery by sternotomy was recruited out of the pre-MICS-era to avoid any unidentified selection bias. Since minimally invasive techniques for the mitral valve came up at our institution beginning from 2016, the control-group, was generated out of the 980 consecutive mitral valve patients fulfilling the inclusion and exclusion criteria between January 2000 and December 2015. This was done to exclude any potential selection bias in the “transition-period” between 2016 and 2018. After a 1:2 propensity score matching the control-group consisted of 320 patients.

### 2.4. Statistical Analysis

The propensity score matching based on a multivariate logistic regression model including age, sex, body-mass-index, EuroScore II and left ventricular ejection fraction. These parameters were chosen to perform a risk and physiognomy-adjustment as the suggested main influencing factors in the decision-making process for or against MICS-MITRAL. According to these parameters, treatment and control-group were 1:2 propensity matched. After matching, all matching factors were counterbalanced between treatment and control-group (Table 1).

Continuous data are expressed as means and standard deviation. Categorial data are displayed as absolute numbers with percentages. For the comparisons of the demographic variables between the two groups, the *t*-test, Mann-Whitney U test or the chi square and Fisher’s exact test was used where appropriate. A *p*-value of <0.05 was considered statistically significant. The statistical analysis was performed using SAS JMP 12.2© (SAS Institute, Cary, NC, USA) and R software, version 4.2.1 (R Foundation for Statistical Computing).

### 2.5. Echocardiographic Assessment

All patients underwent transesophageal echocardiographic examination by an institutionalized core echocardiography laboratory preoperatively and pre discharge transthoracic echocardiographic reevaluation on day 6 after the procedure. The assessment of mitral valve pathologies and their grading were in accordance with the recommendations of the European Association of Cardiovascular Imaging in force at the time.

### 2.6. Surgical Techniques

Regardless of surgical approach, general anesthesia was used in all patients. Intra-operative transesophageal echocardiography was also performed as standard imaging monitoring in all patients.

### 2.7. Treatment Group—Transaxillary Minimally Invasive Mitral Valve Surgery

The patient was placed in supine position with the right side of the chest slightly being elevated by two pillows. The right arm was lifted and placed in a javelin-thrower position and fixed to a surgical table arm support. The airway was intubated with a double-lumen breathing tube allowing single lung ventilation. A temporary transvenous pacing wire was introduced through a percutaneous sheath introducer. The access is performed like described previously [12,13]. In short, a 5 cm skin incision is made in the right anterior axillary line to open the 4th intercostal space. Extracorporeal circulation (ECC) is established through a right femoral access after standard surgical cut-down. After establishing ECC, the pericardium is opened, and pericardial stay sutures are placed. After cross clamping the aorta using a flexible Cosgrove-clamp, crystalloid cardioplegia is administered for cardiac arrest. The left atrium is opened using a left atrial atriotomy in the inter-atrial groove. After that, full exposure of the mitral valve is possible in direct vision using a percutaneous retractor. Specific equipment needed is a headlight, soft tissue retractor and minimally invasive instruments. Figure 1 depicts the intra-operative setup and the postoperative cosmetic result.

### 2.8. Control Group—Mitral Valve Surgery through Full Sternotomy

Full sternotomy cardiac surgery is a well-known procedure. In brief, complete median sternotomy and subsequent pericardial opening were performed in the usual manner. Cardiopulmonary bypass was established by cannulation of the ascending aorta and venous drainage via the superior and inferior vena cava, or two-stage cannulation of the right atrium, depending on the operating surgeon’s preference. Antegrade crystalloid cardioplegia was administered via the ascending aorta. The left ventricular venting line was placed via the right superior pulmonary vein. Atriotomies used were (I) left atrial through the interatrial groove, (II) trans-septal through the right atrium or (III) extended superior trans-septal access as described by Guiraudon et al. [14]. The choice of the access route depended on the surgeon’s preference.

## 3. Results

### 3.1. Patient Baseline Characteristics

After successful propensity matching, the baseline characteristics mainly were counterbalanced between both groups (Table 1 and Table 2). Nonetheless, some residual differences of items being not part of the multivariate regression model were observed:

In the matched data, patients undergoing sternotomy had a significantly higher incidence of arterial hypertension (89.3% vs. 81.3%, *p* = 0.01), dyslipidemia (49.7 vs. 39.4, *p* = 0.03), and preoperative atrial fibrillation (42.8% vs. 33.1%, *p* ≤ 0.001). Furthermore, the calculated creatinine clearance according to the Cockcroft–Gault equation (mL/min) was lower in the Sternotomy group (62.3 ± 23.0 vs. 69.9 ± 20.8, *p* ≤ 0.001). Patients who underwent MICS-MITRAL were more likely to be New York Heart Association (NYHA) class III or IV (65.6% vs. 56.6%, *p* ≤ 0.01) and had a higher incidence of transient ischemic attacks in their medical history (5% vs. 0.6%, *p* ≤ 0.01). Other baseline characteristics did not differ significantly after matching. The baseline characteristics are shown in Table 2.

### 3.2. Mitral Valve Pathology

Regarding MV pathology, more complex vitia in the sense of more degenerative pathologies were treated in the MICS-MITRAL group, meanwhile functional changes were more frequent in the Sternotomy group (*p* = 0.04). Table 3 summarizes MV pathology and echocardiographic findings on admission. According to Carpentier’s classification of mitral valve pathology, type II occurred less frequently in the Sternotomy group (52.6%) than in the MICS-MITRAL group (72.3%; *p* ≤ 0.001), with type IIIa having a higher incidence (14.1% vs. 3.8%; *p* ≤ 0.001; Figure 2).

A higher occurrence of posterior mitral leaflet prolapse (PML) was recorded in the MICS-MITRAL group (62.5% vs. 34.3%; *p* ≤ 0.001). While anterior mitral leaflet (AML) flail was more frequently detected in patients from the Sternotomy group (5.4% vs. 0.6%; *p* ≤ 0.001), posterior mitral leaflet (PML) flail was more frequently evident in the MICS-MITRAL group (22.5% vs. 10.4%; *p* ≤ 0.001). When comparing the leaflet restriction as etiologic cause of mitral valve pathology, the Sternotomy group had a higher proportion of patients with restricted bileaflet motion (14.2% vs. 1.9%; *p* ≤ 0.001). The frequency of annulus dilatation was comparable in both groups (53.8% vs. 50%; *p* = 0.44). Table 3 summarizes the data on mitral valve pathology.

### 3.3. Surgical Procedures Performed on the Mitral Valve

In both groups, mitral valve repair was the dominant procedure and was performed more often than mitral valve replacement. Mitral valve repair was achieved in 64.4% of patients who underwent MICS-MITRAL compared to 71.3% in the Sternotomy group (*p* = 0.31). Replacement after failed repair was also comparable between both treatment groups (Table 4). Among the mitral valve repair procedures, only semirigid rings were implanted in the MICS-MITRAL group. In the Sternotomy group, the proportion of semirigid rings was 90.7% (*p* ≤ 0.001). The reconstructive surgical procedures on AML (0.6% vs. 5.3%; *p* ≤ 0.001) and PML (51.9 vs. 67.4, *p* ≤ 0.001) were less frequently performed in the MICS-MITRAL group. In contrast, the use of artificial cords at the PML was more common in the MICS-MITRAL group. In the MICS-MITRAL group, biological valves were implanted more often in valve replacement procedures (87.7% vs. 72.8%; *p* = 0.04)., with mechanical prosthetic valves via sternotomy being used more frequently (27.2% vs. 12.3%; *p* = 0.04). The skin-to-skin time was not significantly different (MICS-MITRAL 132 ± 31 min vs. Sternotomy 131 ± 46 min; *p* = 0.76). Cardiopulmonary bypass time (CPBT; 88 ± 26.6 vs. 68.7 ± 32.7 min; *p* ≤ 0.001) and aortic cross clamp time (ACCT; 64.4 ± 22.3 vs. 49.7 ± 22.4; *p* ≤ 0.001) were significantly longer in the MICS-MITRAL compared to the Sternotomy group (Figure 3). There were no conversions to sternotomy observed in the MICS-MITRAl group. Table 4 shows the data of surgical procedures performed on the mitral valve.

### 3.4. Concomitant Surgical Procedures

Regarding concomitant surgical procedures, the Sternotomy group had a significantly higher proportion of patients undergoing ablation treatment (34.1% vs. 11.3%, *p* ≤ 0.001). In the MICS-MITRAL group, cryo-ablation was used exclusively. Left atrial appendage occlusion (LAAO) by suture or clip was performed more frequently in the Sternotomy group (25.7% vs. 1.9%, *p* ≤ 0.001). MICS-MITRAL patients were more likely to have concomitant closure of persisting foramen ovale (3.1% vs. 0.3%; *p* = 0.04; Table 5).

### 3.5. Postoperative Morbidity and Mortality

The MICS-MITRAL was characterized by a significantly shorter primary postoperative ventilation time (6.4 ± 6.4 h vs. 15.8 ± 2.4 h, *p* = 0.02). The Sternotomy group had a longer overall ICU stay (78.8 ± 5.8 h vs. 58.8 ± 44.5 h, *p* = 0.05). The intra-operative and postoperative transfusion of packed red blood cells was significantly lower in the MICS-MITRAL group (*p* ≤ 0.001 and *p* = 0.01, respectively). The MICS-MITRAL approach was also associated with significantly lower rate of re-exploration for postoperative bleeding (3.8% vs. 5.6%; *p* = 0.04). Low cardiac output syndrome, defined as prolonged inotropic support for more than 24 h postoperative (any dose), was less frequently observed in the MICS-MITRAL group (16.9% vs. 26.9%, *p* = 0.02). Renal failure with consecutive use of continuous vena-venous hemofiltration was also significantly less prevalent in the MICS-MITRAL group (0.6% vs. 5.9%, *p* ≤ 0.001). In addition, no in-hospital mortality was recorded after minimally invasive surgery, and thus was significantly lower than in the Sternotomy treatment group (0.0% vs. 3.4%, *p* = 0.02). Overall, no significant differences existed among major adverse cardio-cerebral events such as stroke, transient ischemic attack, and myocardial infarction. Further procedural related complications, as respiratory failure (defined as primary postoperative ventilation time ≥ 72 h, re-intubation and/or tracheotomy), overall hospital stay, impaired wound healing, postoperative delirium, permanent pacemaker implantation, new onset atrial fibrillation, and occlusion of the circumflex artery were distributed evenly (Table 6). The graphical overview of the postoperative outcomes is shown in Figure 4.

## 4. Discussion

Inarguably, there exists a strong patient demand for less trauma, less pain, and improved cosmetics, particularly in valve surgery. The best answer to this demand remains part of a lively debate.

Percutaneous techniques can be a possible solution, but due to the limited quality of the achievable results, their appliance remains restricted to inoperable patients [15]. On the other hand, cardiac surgery continuously strives for better minimally invasive techniques [9].

Hereby, the history of mitral valve surgery can be divided into two parts. The early phase, beginning in 1923 when E.C. Cutler performed a first kind of mitral valve surgery, and ending with the pioneering description of standardized pathologies and corresponding reconstruction techniques by Alain Carpentier and at last Patrick Perrier in 1983 and 1997, respectively [9,16,17].

The second phase of evolution in mitral valve surgery dealt with access to the valve. This development, began with Cosgrove and Cohn in the late 1990′s reporting direct vision MICS through a larger thoracotomy, going over a video-assisted technique by Carpentier, and a video-directed method by Chitwood Jr. [18,19,20,21]. Finally, Mohr standardized a technique of video-directed MICS-MITRAL surgery, which finally helped to spread the technique to many centers [11]. The latest advance in MICS-MITRAL surgery is a 3D-non-rib-spreading, full endoscopic approach, leaving robotic approaches on the shelf [10].

Naturally, the development from direct vision to true endoscopic techniques is seen as a (scientifically unproven) continuity of improvement. From that point of view, the transaxillary technique, based on direct vision, could be perceived one step back. Nonetheless, we believe that the transaxillary “single incision—direct vision”-technique has some strikingly strong points.

The price for more video-directed, minimally invasive surgery is an increasingly complex setup and a more and more specialized surgery. Holzhey et al. could demonstrated this fact in their series evaluating learning curves in minimally invasive mitral surgery [11]. They were describing an initial needed caseload of 75 and 125 cases, followed by >1 case per week and surgeon [11]. It must be doubted, that this case load of isolated mitral valve cases is realistic for the vast majority of centers. In this context, it becomes clear, why the pervasion of minimally techniques in mitral valve surgery reaches only 56% in Germany [8]. The technique evaluated in this series, the transaxillary access, is a “single incision—direct vision” approach using a simplified setup without the need for any sophisticated and expensive additional material [12,13]. The skin incision is around 5 cm, which at least is 1 cm to 2 cm more than described in 3D-endoscopic approaches [10,12,13]. Conceding these 2 cm longer skin incisions, transaxillary access, which was performed by three surgeons in this series, obviously provides a more beneficial initial learning curve and a lower needed caseload in the following.

However, there is a difference between the invasiveness of the thoracic access and the invasiveness of the action taken on the mitral valve. The latter conceptually must be the same as “classic” surgery by sternotomy: an extracorporeal circulation is established and there, of course, is a need for temporary cardiac arrest. Correspondingly, several working groups showed that perfusion and aortic cross clamp time are significantly longer in minimally invasive mitral surgery, which might be a result of limited visualization, and unfamiliar hand-eye coordination [1,22,23,24].

We could not confirm the partially excessive mean cross-clamp times of endoscopic approaches, around 120 min and mean perfusion times around 180 min, as reported by the Hamburg group, respectively [10]. In the present series, mean cross-clamp times less than 65 min and mean perfusion times less than 90 min were observed. Nota bene, both times were significantly longer compared to the Sternotomy group, but this did not translate into longer procedure times: In this series, skin-to-skin times around 130 min were observed in the MICS-MITRAL and the Sternotomy group as well. These nearby were dramatically shorter compared to the video-directed operation times between 238 and 277 min, as reported by the Hamburg-group, respectively [10]. The tremendous differences in procedural times could not only be explained by the higher reconstruction rates in the Hamburg series, rather than by the obvious technical advantages of the transaxillary “single incision-direct vision”-approach.

One further concern in MICS-MITRAL surgery regards the feasibility of more complex repair techniques due to the overall limited view of the surgical field compared to conventional surgery. In line with the present literature, the actual study could not see any significant differences—all surgical techniques used in the Sternotomy group were also applicable via the transaxillary access under direct vision in the MICS-MITRAL group [9,10].

It must be noticed, that the repair-rate in the present series was significantly lower compared to the rates reported in some literature [10]. A possible explanation might be the differences and specifics of the underlying pathologies. Potentially the high rate of functional causes of mitral regurgitation and the institutions policy to replace more likely in presence of severe tenting, high probability of SAM-phenomen and ischemic mitral regurge which now is supported by the results given by Acker et al., are possible reasons [25]. Nonetheless, a retrospective elaboration on this was not possible, but for the question of comparing the clinical performance of the analyzed access routes, this should not have a fundamental impact.

Despite all surgical techniques and complex repairs are technically feasible in MICS-MITRAL, it must be concluded that cross-clamp and perfusion times are longer [2,6,7]. The present series suggest, that the transaxillary technique is able to minimize this difference and therefore the additional harm set.

Frankly, less invasive surgical access at least remains more invasive at the actual core part of the procedure. Therefore, the crucial question to be answered is:

Do the clinical advantages of less invasive surgical access outweigh the more invasive surgery at the valve itself?

There exists a significant body of studies elaborating on the clinical performance of MICS-MITRAL surgery. Generally, MICS-mitral surgery seems to be safe, but more time-consuming [1,2,3,7,24,26]. To date, there exists some evidence, that the “soft skills” of MICS-MITRAL surgery by means of bleeding, need for transfusion and postoperative pain are beneficial [9,10]. In line with the literature, the present study likewise observed shorter ventilation times, shorter ICU stay, less transfusions, and less re-exploration for bleeding in the MICS-MITRAL group compared to the sternotomy patients [2,24,27,28].

A particular phenomenon in MICS-MITRAL seems to be the unilateral pulmonary edema during the postoperative period, or impaired pulmonary function due to capillary leakage or single lung ventilation [24,29]. For example, Wang et al. reported prolonged extubation in the minimally invasive group. They discussed a possible systemic inflammatory response, consecutive pulmonary capillary leakage and pulmonary edema due to prolonged CPBT leading to transient worsening of pulmonary function as the cause. Another explanation was the use of double-lumen tube, which may have led to development of segmental or lobar atelectasis [24]. Contradictory, in the present study, we could not reproduce these adverse results—quite the contrary, the MICS-MITRAL had shorter ventilation times and less postoperative respiratory failure in this series.

Against the background, that de-airing naturally is an issue in minimally invasive, the stroke rate remarkably did not differ significantly between MICS-MITRAL and sternotomy in the present series.

Remarkably, present literature could not demonstrate neither beneficial nor harming effect with the regard to the only real “hard” endpoint “mortality” of MICS-MITRAL surgery [2,3,5,6,7]. On the contrary, the present series proved significantly better survival of the transaxillary “single incision-direct vision” access compared to classic sternotomy approach—to our best knowledge for the first time in literature. Hereby, it must be highlighted, that the mortality rate of the compared Sternotomy group in this series is absolutely in line with the results reported in literature [8].

## 5. Conclusions

After successful 1:2 propensity matching, the transaxillary “single incision-direct vision” approach is characterized by equal procedure times, shorter ventilation times, shorter ICU-stay, less bleeding, less need for transfusion and less re-exploration for bleeding. The quality of the mitral valve surgery itself was comparable to the sternotomy cases. There were no significant differences in postoperative stroke rate or transient neurologic symptoms. Finally, transaxillary MICS-MITRAL can be performed with at least the same safety and in a comparable time frame as conventional mitral surgery by sternotomy. Obvious advantages are the cosmetic results and the negligible physical limitations due to the preservation of the sternum and the ribs.

## 6. Limitations

This study has several inherent limitations. The principal limitation is the retrospective character of this analytical comparison, in that the bias due to the retrospective approach cannot be avoided. Although propensity score matching was performed to minimize allocation bias between the two treatment groups, not all measured confounders could be accounted for. A particular concern might be that since the matching parameters primarily were chosen for risk and physiognomy-adjustment, the matching for the remaining baseline and echo characteristics was not perfect (e.g., atrial fibrillation, creatinine clearance). Additionally, the differing types of repair as well as cause for regurgitation might have caused an undetected bias. Moreover, our study is the result of retrospective data collection at a single center, and therefore may be influenced by subjective factors such as surgical experience and peri-operative management. Finally, the lack of follow-up data must be mentioned.

## Figures and Tables

**Figure 1 medicina-58-01850-f001:**
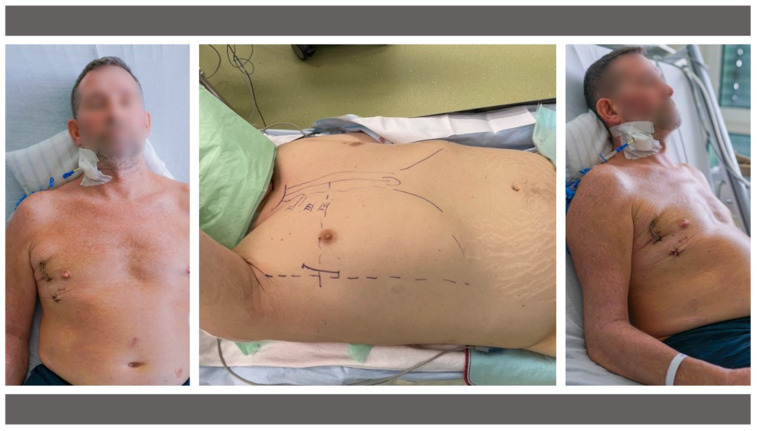
Intra-operative setup (**middle**) and the postoperative cosmetic result (**left**/**right**) of transaxillary minimally invasive mitral valve surgery on postoperative day 2.

**Figure 2 medicina-58-01850-f002:**
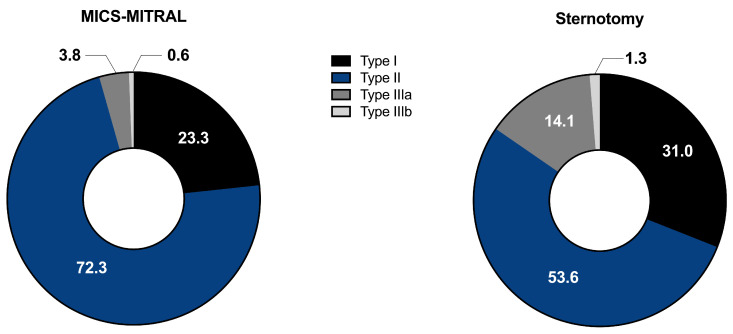
Intra-operative setup (**middle**) and the postoperative cosmetic result (**left**/**right**) of transaxillary minimally invasive mitral valve surgery on postoperative day 2.

**Figure 3 medicina-58-01850-f003:**
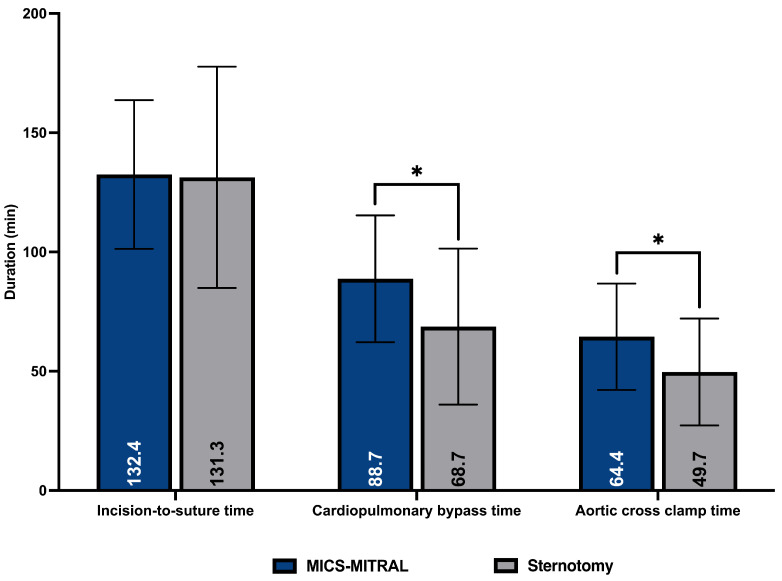
Procedural times comparing MICS-MITRAL and Sternotomy. Note: *, *p* < 0.05.

**Figure 4 medicina-58-01850-f004:**
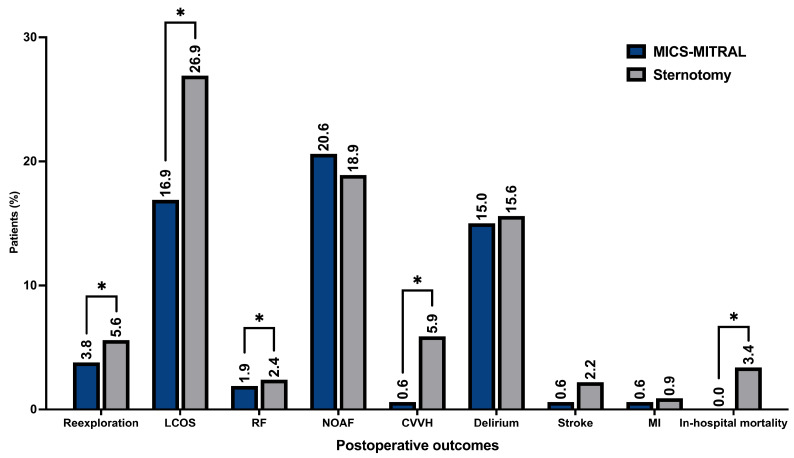
Postoperative clinical performance of MICS-MITRAL and conventionally treated by sternotomy. Note: * *p* ≤ 0.05; LCOS…low cardiac output syndrome defined as prolonged inotropic support > 24 h (any dose); RF…respiratory failure defined as primary postoperative ventilation time ≥ 72 h, re-intubation and/or tracheotomy; NOAF…new onset atrial fibrillation; CVVH…continuous vena-venous hemofiltration; MI…myocardial infarction.

**Table 1 medicina-58-01850-t001:** Univariate analysis of balanced matching parameters after 1:2 propensity score matching.

	Propensity Matched Cohort (*n* = 480)		
	MICS-MITRAL (*n* = 160)	Sternotomy (*n* = 320)	*p*-Value
Age (years), mean ± SD	66.7 ± 11.4	66.5 ± 10.2	0.89
Sex (female), *n* (%)	70 (43.8)	164 (51.3)	0.15
BMI (kg/m^2^), mean ± SD	25.9 ± 3.3	26.2 ± 3.4	0.32
EuroSCORE II (%), mean ± SD	1.9 ± 2.3	2.0 ± 0.9	0.58
LVEF (%), mean ± SD	56.1 ± 1.1	54.9 ± 0.8	0.37

Abbreviations: MICS-MITRAL, minimally invasive mitral valve surgery; SD, standard deviation; BMI, body mass index; LVEF, left ventricular ejection fraction.

**Table 2 medicina-58-01850-t002:** Baseline characteristics.

	Propensity Matched Cohort (*N* = 480)		
	MICS-MITRAL (*n* = 160)	Sternotomy (*n* = 320)	*p*-Value
**Age (years), mean ± SD**	66.7 ± 11.4	66.5 ± 10.2	0.89
**Sex (male), *n* (%)**	70 (43.8%)	164 (51.3)	0.15
**BMI (kg/m^2^), mean ± SD**	25.9 ± 3.3	26.2 ± 3.4	0.32
**Diabetes mellitus, *n* (%)**	25 (15.6)	69 (21.6)	0.11
**Coronary artery disease, *n* (%)**	32 (20.0)	89 (27.8)	0.06
**LVEF (%), mean ± SD**	56.1 ± 1.1	54.9 ± 0.8	0.37
**COPD, *n* (%)**	11 (6.9)	29 (9.1)	0.41
**Pulmonary arterial hypertension, *n* (%)**	93 (58.9)	193 (60.3)	0.19
**Hemodialysis, *n* (%)**	1 (0.6)	3 (0.9)	0.72
**Creatinine Clearance** **(mL/min), mean ± SD**	69.9 ± 20.8	62.3 ± 23.0	** *≤0.001 *** **
**Peripherial artery disease, *n* (%)**	1 (0.6)	8 (2.5)	0.15
**Carotid artery stenosis > 50%, *n* (%)**	6 (3.4)	8 (2.5)	0.45
**h/o TIA, *n* (%)**	8 (5.0)	2 (0.6)	** *≤0.01 *** **
**h/o Ischemic stroke, *n* (%)**	39 (4.0)	23 (6.0)	0.14
**Atrial fibrillation, *n* (%)**	53 (33.1)	137 (42.8)	** *≤0.001 *** **
**Pacemaker, *n* (%)**	9 (5.6)	31 (9.7)	0.13
**Smoker status, *n* (%)**	25 (15.6)	31 (9.8)	0.06
**NYHA class III or IV, *n* (%)**	105 (65.6)	181 (56.6)	** *≤0.01 *** **
**Frailty, *n* (%)**	7 (4.4)	9 (2.8)	0.40
**EuroSCORE II (%), mean ± SD**	1.9 ± 2.3	2.0 ± 0.9	0.58

Note: Bold values indicate statistical significance: *, *p* ≤ 0.05; **, *p* ≤ 0.01. Abbreviations: MICS-MITRAL, minimally invasive mitral valve surgery; SD, standard deviation; BMI, body mass index; LVEF, left ventricular ejection fraction; COPD, chronic obstructive pulmonary disease; h/o history of; TIA, transient ischemic attack; NYHA, New York Heart Association.

**Table 3 medicina-58-01850-t003:** Mitral valve pathology and echocardiographic parameters.

	Mitral Valve Pathology				
	MICS-MITRAL		Sternotomy		*p*-Value
	*n/N*	%	*n/N*	%	
**Degenerative vs. functional**	***N* = 160**		***N* = 302**		
Degenerative, *n*	116/160	72.5	192/302	63.6	** *0.04 ** **
Functional, *n*	36/160	22.5	96/302	31.8	
**Carpentier classification**	***N* = 159**		***N* = 313**		
Type I, *n*	37/159	23.3	97/313	31	** *≤0.001 *** **
Type II, *n*	115/159	72.3	168/313	53.6	
Type IIIa, *n*	6/159	3.8	44/313	14.1	
Type IIIb, *n*	1/159	0.63	4/313	1.3	
**Prolapse site**	***N* = 160**		***N* = 318**		
AML, *n*	9/160	5.6	38/318	12	** *≤0.001 *** **
PML, *n*	100/160	62.5	109/318	34.3	
Bileaflet, *n*	4/160	2.5	19/318	6	
**Flail**	***N* = 160**		***N* = 318**		
No flail, *n*	123/160	76.9	50/328	84.2	** *≤0.001 *** **
AML, *n*	1/160	0.6	17/318	5.4	
PML, *n*	36/160	22.5	33/318	10.4	
**Restricted leaflet motion**	***N* = 160**		***N* = 318**		
AML, *n*	4/160	2.5	4/318	1.3	** *≤0.001 *** **
PML, *n*	11/160	6	19/318	6	
Bileaflet, *n*	3/160	1.9	45/318	14.2	
**Annulus dilatation, *n***	***N* = 160**		***N* = 318**		
	86/160	53.8	159/318	50	0.44

Note: Bold values indicate statistical significance: *, *p* ≤ 0.05; **, *p* ≤ 0.01. Abbreviations: MICS-MITRAL, minimally invasive mitral valve surgery; AML, anterior mitral leaflet; PML, posterior mitral leaflet.

**Table 4 medicina-58-01850-t004:** Data of surgical procedures performed on the mitral valve.

	Surgical Procedures				
	MICS-MITRAL		Sternotomy		*p*-Value
	*n/N*	%	*n/N*	%	
**Type of surgery**	***N* = 160**		***N* = 320**		
Repair, *n*	103/160	64.4	228/320	71.3	0.31
Replacement, *n*	51/160	35.6	82/320	25.6	
Failed repair, *n*	6/160	3.75	10/320	3.1	
**Type of Annuloplasty Ring**	***N* = 101**		***N* = 225**		
Flexible, *n*	0/101	9	16/225	7.1	** *≤0.001 *** **
Rigid, *n*	0/101	0.0	5/225	2.2	
Semirigid, *n*	101/101	100.0	204/225	90.7	
**MV repair site**	***N* = 160**		***N* = 319**		
AML, *n*	1/160	0.6	17/319	5.3	** *≤0.001 *** **
PML, *n*	83/160	51.9	215/319	67.4	
Bileaflet, *n*	3/160	1.9	8/319	2.5	
**Prosthesis classification**	***N* = 57**		***N* = 92**		
Mechanical, *n*	7/57	12.3	25/92	27.2	** *0.04 ** **
Biological, *n*	50/57	87.7	67/92	72.8	
**Resection**	***N* = 160**		***N* = 319**		
No resection, *n*	137/160	85.6	247/319	77.4	0.09
AML, *n*	0/160	0.0	3/319	0.9	
PML, *n*	23/160	14.4	65/319	20.4	
Bileaflet, *n*	0/160	0.0	4/319	1.3	
**Artificial chords**	***N* = 160**		***N* = 319**		
No artificial chords, *n*	102/160	63.8	284/319	89	** *≤0.001 *** **
AML, *n*	1/160	0.6	13/319	4.1	
PML, *n*	57/160	35.6	21/319	6.6	
Bileaflet, *n*	0/160	0.0	1/319	0.3	
**Edge-to-edge repair**	***N* = 160**		***N* = 319**		
No edge-to-edge repair, *n*	156/160	97.5	315/319	98.8	0.45
Edge-to-edge repair, *n*	4/160	2.5	4/319	1.3	
**Conversion to sternotomy**	***N* = 160**		***N* = 320**		
	0/160	0.0	0/320	0.0	≥0.99
**Skin-to-Skin-Time (min), mean ± SD**	132.0 ± 31.0		131.0 ± 46.0		0.76
**Perfusion time, mean ± SD**	88.7 ± 26.6		68.7 ± 32.7		** *≤0.001 *** **
**X-clamp time, mean ± SD**	64.4 ± 22.3		49.7 ± 22.4		** *≤0.001 *** **

Note: Bold values indicate statistical significance: *, *p* ≤ 0.05; **, *p* ≤ 0.01. Abbreviations: MICS-MITRAL, minimally invasive mitral valve surgery; RAFR, replacement after failed repair; ARC, annuloplasty ring classification; ART, annuloplasty ring type; SJM, St. Jude Medical; MDT, Medtronic; AML, anterior mitral leaflet; PML, posterior mitral leaflet.

**Table 5 medicina-58-01850-t005:** Concomitant surgical procedures.

	Propensity Matched Cohort (*N* = 480)				
	MICS-MITRAL		Sternotomy		*p*-Value
	*n/N*	%	*n/N*	%	
**AF ablation**	***N* = 160**		***N* = 320**		
No AF ablation, *n*	142/160	88.7	211/320	65.9	** *≤0.001 *** **
AF ablation, *n*	18/160	11.3	109/320	34.1	
**AF surgery technique**	***N* = 18**		***N* = 109**		
Radiofrequency, *n*	0/18	0.0	47/109	43.1	** *≤0.001 *** **
Microwave, *n*	0/18	0.0	24/109	22	
Cryoablation, *n*	18/18	100.0	38/109	34.86	
**LAAO**	***N* = 160**		***N* = 319**		
No LAAO	157/160	98.1	237/319	74.3	** *≤0.001 *** **
Clip/Suture	3/160	1.9	82/319	25.7	
**Others**	***N* = 160**		***N* = 320**		
PFO closure, *n*	5/160	3.1	1/320	0.3	** *0.04 ** **
Tumor resection, *n*	1/160	0.6	0/320	0.0	
PDA closure, *n*	0/160	0.0	1/320	0.3	

Note: Bold values indicate statistical significance: *, *p* ≤ 0.05; **, *p* ≤ 0.01. Abbreviations: MICS-MITRAL, minimally invasive mitral valve surgery; AF, atrial fibrillation; LAAO, left atrial appendage occlusion; PFO, persisting foramen ovale; PDA, persisting ductus arteriosus (Botalli).

**Table 6 medicina-58-01850-t006:** Postoperative procedural-related complications.

	Propensity Matched Cohort (*N* = 480)		
	MICS-MITRAL (*n* = 160)	Sternotomy (*n* = 320)	*p*-Value
Ventilation time (hours), mean ± SD	6.4 ± 6.4	15.8 ± 2.4	** *0.02 ** **
Respiratory failure ^†^, *n* (%)	3 (1.9)	9 (2.4)	0.52
ICU stay (hours), mean ± SD	58.8 ± 44.5	78.8 ± 5.8	** *0.05 ** **
Hospital stay, mean ± SD	11.7 ± 5.2	12.5 ± 6.7	0.16
Intraop. transfusion (PRBC), mean ± SD	0.1 ± 0.5	0.5 ± 1.1	** *≤0.001 *** **
Postop. transfusion (PRBC), mean ± SD	0.3 ± 1.3	1.3 ± 5.0	** *0.01 ** **
CVVHD, *n* (%)	1 (0.6)	19 (5.9)	** *≤0.001 *** **
Conversion to full sternotomy, *n* (%)	0 (0.0)	0 (0.0)	≥0.99
Re-exploration for bleeding, *n* (%)	6 (3.8)	18 (5.6)	** *0.04 ** **
Impaired wound healing, *n* (%)	4 (2.5)	16 (5.0)	0.2
Delirium, *n* (%)	24 (15.0)	50 (15.6)	0.89
Stroke, *n* (%)	1 (0.6)	7 (2.2)	0.47
TIA, *n* (%)	1 (0.6)	3 (0.9)	0.17
Permanent pacemaker implantation, *n* (%)	8 (5.0)	7 (2.2)	0.10
New Onset AF, *n* (%)	33 (20.6)	60 (18.9)	0.71
Prolonged inotropic support > 24 h (any dose), *n* (%)	27 (16.9)	86 (26.9)	** *0.02 ** **
RCX occlusion, *n* (%)	0 (0.0)	1 (0.3)	≥0.99
In-hospital mortality *n* (%)	0 (0.0)	11 (3.4)	** *0.02 ** **

Note: Bold values indicate statistical significance: *, *p* ≤ 0.05; **, *p* ≤ 0.01; †, defined as primary postoperative ventilation time ≥ 72 h, re-intubation and/or tracheotomy; Abbreviations: AF, Atrial fibrillation; MICS-MITRAL, minimally invasive mitral valve surgery; ICU, intensive care unit; CVVHD, consecutive renal failure needing continuous veno-venous hemofiltration; TIA, transient ischemic attack; RCX, circumflex artery.

## Data Availability

The data presented in this study are available on request from the corresponding author. The data are not publicly available due to ethical regulations.

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
