# Peer review of "Safety and Efficacy of the Transaxillary Access for Minimally Invasive Mitral Valve Surgery—A Propensity Matched Competitive Analysis"

_medicina, 2022, doi:10.3390/medicina58121850_

Round 1

Reviewer 1 Report

This study compares the outcomes of single-institutional consecutive patients undergoing transaxillary access for minimally invasive mitral valve surgery (MICS-MITRAL) and historical patients undergoing sternotomy approach. The two groups were 1:2 propensity matched. The authors should be congratulated on their excellent results including zero mortality and low complications rates in the MICS-MITRAL group. However, whereas the transaxillary group has almost perfect outcomes, their conclusion that MICS-MITRAL provides superior clinical outcomes with significantly better hospital survival may be questionable.

First, their propensity match included only 5 variables, which made the MICS-MITRAL and Sternotomy groups remain different. For example, the Sternotomy group had significantly lower creatinine clearance, higher atrial fibrillation rate, etc. It would make more sense to include these variables in the propensity match methodology.

Second, the control group comes from historical patients prior to the introduction of the transaxillary access. While their purpose to avoid selection bias seems reasonable, readers would be interested in knowing how many patients underwent mitral surgery via sternotomy between 2019 and 2021 when the transaxillary access was performed. If the number of patients undergoing sternotomy turns out to be substantial, selection bias would remain obvious.

Nevertheless, their conclusion that MICS-MITRAL is associated with significantly survival benefit seems provocative and should be carefully revisited.

Finally, the manuscript contains multiple typos. On Page 10, line 241 the percentage of LAAO by suture or clip should be 1.9% to be consistent with Table 5. The PFO closure rate should be 3.1% (line 242). In Figure 4, the abbreviations were not spelled out. The RF (respiratory failure?) rate and associated p valve for MICS-MITRAL are apparently incorrect, and not consistent with Table 6.

Author Response

Dear Reviewer #1,

Thank you very much for your constructive review, which will help to significantly improve the manuscript. We hopefully addressed each of your comments as adequately as possible. In the manuscript, the changes made are highlighted red. Please find below the point-to-point- response:

#1             “However, whereas the transaxillary group has almost perfect outcomes, their conclusion that MICS-MITRAL provides superior clinical outcomes with significantly better hospital survival may be questionable.”

--> We changed our somehow tendentious conclusion into: The transaxillary access for MICS-MITRAL provides perfect cosmetics and excellent clinical outcomes. It can be performed at least as safe and in the same time frame as conventional mitral surgery by sternotomy. In the manuscript we added “Finally, transaxillary MICS-MITRAL can be performed with at least the same safety and in a comparable time frame as conventional mitral surgery by sternotomy. Obvious advantages are the cosmetic result and the negligible physical limitations due to preservation of the sternum and the ribs.“

#2             First, their propensity match included only 5 variables, which made the MICS-MITRAL and Sternotomy groups remain different. For example, the Sternotomy group had significantly lower creatinine clearance, higher atrial fibrillation rate, etc. It would make more sense to include these variables in the propensity match methodology.

--> This is a point we actually had a lot of discussion within our group. Finally, we decided for some kind of risk-and physiognomy-adjustment as the main preoperative factors (at least for us) for historically deciding against MICS-MITRAL. This was done in favor of a larger number of included patients. But you are absolutely right, this did not include the whole bias. Interestingly we thought that by using the EuroSCORE II (which includes GFR), renal function will be counterbalanced – which actually did not perfectly happen (maybe because of the weight in the EuroSCORE regression modell). Accordingly we put a corresponding paragraph in the Limitations-section (“A particular concern might be, that – since the matching parameters primarily were chosen for risk and physiognomy-adjustment – the matching for the remaining baseline characteristics was not perfect (e.g. atrial fibrillation, creatinine clearance”) and better explained the choice of matching variables in the Methods part. (These parameters were chosen to perform a risk and physiognomy-adjustment as the suggested main influencing factors in the decision-making process for or against MICS-MITRAL.).

#3             Second, the control group comes from historical patients prior to the introduction of the transaxillary access. While their purpose to avoid selection bias seems reasonable, readers would be interested in knowing how many patients underwent mitral surgery via sternotomy between 2019 and 2021 when the transaxillary access was performed. If the number of patients undergoing sternotomy turns out to be substantial, selection bias would remain obvious.

--> You are right, that is an important information. We added this correspondingly in the Methods section (During the period between 2019 and 2021 a total of 168 patients underwent isolated mitral surgery according to the inclusion criteria. Hence, the corresponding MICS-proportion was 95.2%. Eight patients (4.8%) were not done transaxillary, due to extensive annular calcifications with conceivable need for decalcification and possible need for posterior mitral patch annuloplasty) and “This was done to exclude any potential selection bias in the “transition-period” between 2016 and 2018.”

#4             Nevertheless, their conclusion that MICS-MITRAL is associated with significantly survival benefit seems provocative and should be carefully revisited.

--> We changed this as mentioned under #1

#5             Finally, the manuscript contains multiple typos.

On Page 10, line 241 the percentage of LAAO by suture or clip should be 1.9% to be consistent with Table 5. --> corrected

The PFO closure rate should be 3.1% (line 242). --> corrected

In Figure 4, the abbreviations were not spelled out. The RF (respiratory failure?) rate and associated p valve for MICS-MITRAL are apparently incorrect, and not consistent with Table 6. --> we added a figure legend and corrected figure 4

Reviewer 2 Report

Would recommend editing for grammar/English language and style

From the propensity matching, it seems that the control group is not similar to the minimally invasive group in terms of reason for regurgitation, type of carpentier type, type of repair etc, ablation or not and occlusion of LAA.  It’s hard to compare then the postoperative results of prolonged intubation, mortality etc if they aren’t matched with similar type of repair, cause for regurgitation etc. 

Author Response

Dear Reviewer #2,

Thank you very much for your constructive review, which helped us to improve the manuscript. We hopefully addressed each of your comments as adequately as possible. In the manuscript, the changes made are highlighted red. Please find below the point-to-point- response:

#1             From the propensity matching, it seems that the control group is not similar to the minimally invasive group in terms of reason for regurgitation, type of carpentier type, type of repair etc, ablation or not and occlusion of LAA. It’s hard to compare then the postoperative results of prolonged intubation, mortality etc. if they aren’t matched with similar type of repair, cause for regurgitation etc. 

--> This is a point we actually did discuss within our group. Before starting the data analysis we decided for some kind of risk-and physiognomy-adjustment as the main preoperative factors (at least for us) for historically deciding against MICS-MITRAL. We did this in favor of a higher number of included patients and we hoped to rule out the bias for choosing or not choosing a minimally invasive access. But we absolutely agree that this might also have caused some undetected bias. Accordingly, we put a corresponding paragraph in the Limitations-section (“A particular concern might be, that – since the matching parameters primarily were chosen for risk and physiognomy-adjustment – the matching for the remaining baseline and echo characteristics was not perfect (e.g. atrial fibrillation, creatinine clearance). Additionally, the differing type of repair as well as cause for regurgitation might have caused an undetected bias”) and better explained the choice of matching variables in the Methods part. (These parameters were chosen to perform a risk and physiognomy-adjustment as the suggested main influencing factors in the decision-making process for or against MICS-MITRAL.).

#2             It’s hard to compare then the postoperative results of prolonged intubation, mortality etc. …

--> Reviewer #1 had comparable concerns concerning our “hard” conclusion. Therefor we changed our somehow tendentious conclusion into: The transaxillary access for MICS-MITRAL provides perfect cosmetics and excellent clinical outcomes. It can be performed at least as safe and in the same time frame as conventional mitral surgery by sternotomy. In the manuscript we added “Finally, transaxillary MICS-MITRAL can be performed with at least the same safety and in a comparable time frame as conventional mitral surgery by sternotomy. Obvious advantages are the cosmetic result and the negligible physical limitations due to preservation of the sternum and the ribs“
